# Prior salpingectomy impairs the retrieved oocyte number in *in vitro* fertilization cycles of women under 35 years old without optimal ovarian reserve

**Cheng-Yu Ho**[1,2], **Yu-Yuan Chang**[3], **Yu-Hung Lin**[1], **Mei-Jou Chen**[2,3,4]*

1 Department of Obstetrics and Gynecology, Shin Kong Wu Huo-Shih Memorial Hospital, Taipei, Taiwan,
2 Graduate Institute of Clinical Medicine, College of Medicine, National Taiwan University, Taipei, Taiwan,
3 Department of Obstetrics and Gynecology, College of Medicine and National Taiwan University Hospital, Taipei, Taiwan, 4 Livia Shang-Yu Wan Chair Professor of Obstetrics and Gynecology, National Taiwan University, Taipei, Taiwan

* mjchen04@ntu.edu.tw

**Data Availability Statement:** Data cannot be shared publicly because of containing potentially identifying or sensitive patient information. We have provided minimal data set, which are fully

## Abstract

### Study objective

The impairment of the ovarian response in *in vitro* fertilization (IVF) cycles after salpingectomy remains contentious. Therefore, we investigated whether a history of salpingectomy affects the number of oocytes retrieved in women undergoing IVF in comparison with the number in women without underlying tubal disease.

### Design

Case–control study (Canadian Task Force Classification II-2).

### Setting

A tertiary hospital–affiliated fertility center.

### Patients

Fifty-four women aged <35 years with a history of salpingectomy and 59 age-matched women without tubal disease.

### Interventions

Gonadotropin-releasing hormone antagonist protocol for controlled ovarian stimulation and transvaginal oocyte retrieval.

### Measurements and main results

The antral follicle count (AFC), anti-Müllerian hormone (AMH) levels, and the number of retrieved oocytes were significantly lower in women with prior salpingectomy than in women without tubal disease. Day-3 follicle-stimulating hormone (FSH) levels, total gonadotropin

anonymized, in the supporting information files.
Data are available from the Institutional Review
Board of Shin Kong Wu Huo-Shih Memorial
Hospital.(contact via email: IRB@ms.skh.org.tw.)
for researchers who meet the criteria for access to
confidential data.

**Funding:** Our study was supported by Ministry of
Science and Technology in Taiwan. (Award
number: MOST 109-2314-B002-125-MY3) The
funders had no role in study design, data collection
and analysis, decision to publish, or preparation of
the manuscript. The authors received no specific
funding for this work.

**Competing interests:** The authors have declared
that no competing interests exist.

dosage, and stimulation days did not significantly differ between the groups. The indications of salpingectomy (i.e., hydrosalpinx and ectopic pregnancy) did not differ significantly in terms of ovarian response or reserve among women with salpingectomy history. A history of salpingectomy and other factors related to ovarian response in IVF, such as age, AMH, AFC, day-3 FSH, and total gonadotropin dose, were significantly correlated with the number of oocytes retrieved by univariate regression analysis. In the multivariate-adjusted model after controlling all the above-mentioned variables, only AFC and AMH levels continued to exhibit significant associations with the number of retrieved oocytes. In a subgroup analysis, the negative impact of prior salpingectomy on the number of retrieved oocytes was especially significant in women with suboptimal ovarian reserves (defined as AMH < 4 ng/mL), regardless of the indication of salpingectomy or whether salpingectomy was bilateral or unilateral.

## Conclusion

A negative effect on the number of retrieved oocytes in the subsequent IVF cycle after salpingectomy is more likely in women aged <35 years with suboptimal ovarian reserve. Nevertheless, postsurgical AMH and AFC levels still possess a more direct predictive value on ovarian response than the history of salpingectomy.

## Introduction

Tubal factor infertility is among the most common causes of female infertility, affecting approximately 30%–35% of women with infertility [1]. Tubal factor infertility can be caused by acute salpingitis, endometriosis, and surgical adhesion, which result in chronic inflammation and distal tubal adhesive disease. Women who require tubal surgery for various conditions tend to select salpingectomy over salpingostomy or proximal tubal ligation to reduce the risk of future epithelial ovarian cancer, incomplete treatment, and reoperation due to postsurgical adhesion or persistent disease [2].

A hydrosalpinx occurs when a blocked fallopian tube fills with fluid, and it is a common disease in women who have undergone *in vitro* fertilization (IVF) or embryo transfer. Previous *in vivo* and *in vitro* studies have revealed that the content of hydrosalpinx fluid may be toxic to an embryo and adversely affect subsequent embryo development and implantation [3–6]. Hydrosalpinges are reportedly associated with a 50% lower chance of implantation and a two-fold increased risk of pregnancy loss in IVF pregnancies [7]. Studies have suggested that prophylactic salpingectomy or proximal tubal obstruction for hydrosalpinges prior to IVF helps increase the ongoing pregnancy and live birth rates [2, 8–11]. However, because of the profound anastomosis of the blood supply between the ovary and fallopian tubes, salpingectomy and other tubal surgeries have the potential to adversely affect fertility and ovarian reserve, causing concern for clinicians and patients [2]. Nevertheless, the data on ovarian response and ovarian reserve after salpingectomy have often been contradictory [2, 12, 13].

Ectopic tubal pregnancy is associated with chronic salpingitis, and typical treatment includes salpingectomy, especially when evidence of rupture is present [14]. Compared with salpingostomy, salpingectomy may be superior in eliminating the risks of persistent and repeat ectopic pregnancies and reducing the number of hospital visits and frequency of blood sampling [14, 15]. In addition to the concern regarding ovarian damage in patients receiving

salpingectomy for hydrosalpinges, many researchers have investigated whether disruptions to the descending vasculature along with the removed fallopian tube could impair ovarian function in ectopic pregnancy; however, disparate results have been obtained [15, 16].

A randomized controlled trial of women with hydrosalpinges [17] indicated lower ovarian reserve, lower numbers of retrieved oocytes, and higher gonadotropin stimulation dosages in patients who underwent salpingectomy compared with those who underwent proximal tubal occlusion. An experimental study revealed that total salpingectomy in rats leads to more significant damage than proximal tubal occlusion in ovarian histopathology and the cholinergic system [18]. Another study also indicated that the unilateral total salpingectomy procedure can be detrimental to ipsilateral ovarian tissue as a result of ischemia–reperfusion injury [19].

Nonetheless, most meta-analyses have been unable to demonstrate a significant detrimental effect of salpingectomy on ovarian reserve and ovarian response in IVF, regardless of the indications of salpingectomy [2, 8–12, 15]. Moreover, studies investigating the effect of salpingectomy on ovarian reserve have typically been designed to compare the surrogate markers of the ovarian reserve or ovarian response of IVF patients before and after surgery within a short period or have limited the study population to patients with tubal disease. Furthermore, most of these studies have not controlled for the ovarian stimulation protocol.

In a study comparing women with infertility with and without history of salpingectomy for known tubal disease [13], history of salpingectomy was associated with a lower antral follicle count (AFC) in the women with infertility who were 35–39 years old. The study did not report a reduction of AFC related to a history of salpingectomy in women under 35 years old, implying that decline in age-related ovarian reserve may contribute to the impairment of ovarian reserve in women with a history of salpingectomy. However, the study did not include a sub-group analysis to investigate whether the effect of salpingectomy was more pronounced in women under the age of 35 years with suboptimal ovarian reserve. In the present study, we (1) investigated whether history of salpingectomy affected the number of oocytes in women undergoing IVF compared with women without underlying tubal disease by using conventional hysterosalpingographic evaluation and (2) examined whether this effect differed depending on ovarian reserve.

The outcomes considered in this study were the indication of salpingectomy, bilateral versus unilateral salpingectomy, and distinct levels of baseline ovarian reserve, which may have influenced the effect of salpingectomy on the number of retrieved oocytes.

## Materials and methods

### Setting and design

This retrospective case–control study was approved by the Institutional Review Board of Shin Kong Wu Huo-Shih Memorial Hospital (Approval number: 20210704R). All clinical data were fully anonymized before we accessed them and the inform consent was waived because of the retrospective nature.

The research was conducted in the Infertility Center of the Department of Obstetrics and Gynecology at Shin Kong Wu Huo-Shih Memorial Hospital from January 2012 through December 2019. During this period, 1096 patients aged under 35 years received IVF treatment in our center, including 102 patients with a history of salpingectomy. For the final analysis, the study group consisted of 54 participants after women who did not meet our inclusion criteria or lacked complete surgical records were excluded. Another 59 participants were randomly selected as age-matched controls (by 3-year age strata) from a consecutive series of 606 women with conventional hysterosalpingography results; these participants received IVF because of

male fertility factors or unexplained infertility. For all the recruited participants, the data for controlled ovarian stimulations were only collected from the first IVF cycle.

The study group was further divided into subgroups according to the indication for salpingectomy (hydrosalpinx or ectopic pregnancy) and the type of salpingectomy (bilateral or unilateral). The inclusion criteria of all the enrolled control and study participants were as follows: (1) aged 20–34 years, (2) had a regular menstrual cycle, and (3) treated with a gonadotropin-releasing hormone (GnRH) antagonist protocol. Patients were excluded if they had a history of ovarian surgery or diseases that affect ovarian function, namely endometriosis, ovarian tumors, polycystic ovarian syndrome, and autoimmune disease. Patients were excluded from the control group if tubal factors were present.

## Salpingectomy

All the enrolled patients who required tubal surgery received laparoscopic salpingectomy from an experienced gynecologic surgeon who endeavored to incise the smallest possible area within the mesosalpinges. We excluded patients who received segmental salpingectomy for severe pelvic adhesion or other inoperable conditions.

## Controlled ovarian stimulation and outcome measures

All the enrolled patients were treated with an individualized GnRH antagonist protocol [20] for controlled ovarian stimulation before oocyte retrieval, which was performed as described previously. In brief, exogenous recombinant follicle-stimulating hormone (rFSH, Gonal F; Merck Serono, Germany) was administered at a dose of 150–375 IU/day depending on the patient's age, body mass index (BMI), ovarian reserve, and response. Serial transvaginal ultrasound scans and concurrent serum luteinizing hormone, estradiol (E2), and progesterone levels were used to assess the ovarian response of all the enrolled patients and adjust the gonadotrophin dosage accordingly. The GnRH antagonist protocol (Cetrotide; Merk Serono, Geneva, Switzerland) was commenced on stimulation day 6 or when the dominant follicle reached 14 mm, whichever occurred first. Administration of one dose of 6500 IU hCG (Ovidrel; Merck Serono) or two 0.1 mg doses of triptorelin (Decapeptyl; Ferring Pharmaceuticals) was performed for ovulation trigger when two or more follicles reached 18 mm in diameter, or three follicles reached 17 mm in diameter. Oocyte retrieval was performed 34–36 hours after triggering.

The number of retrieved oocytes, total administered dose of gonadotropin, and total days of ovarian stimulation were recorded. The biomarkers anti-Müllerian hormone (AMH), day-3 FSH, and AFC on menstrual cycle day 2 or 3 were also measured before patients entered the IVF cycle to predict ovarian reserve. Subgroup analyses were conducted to evaluate the association between the number of retrieved oocytes and salpingectomy in women with infertility undergoing IVF with various baseline ovarian reserves. The cutoff value for AFC (12 follicles) was based on the calculated median AFC of all enrolled patients, and the cutoff value for AMH (4 ng/mL) was based on the reported 75th percentile of AMH levels of generally healthy Chinese women aged 35 years. The AMH levels between 35 to 36 year-old Chinese women lies between 4.229 to 3.973 ng/mL in the 75th percentile [21].

## Statistical analysis

Numeric variables are presented as untransformed means ± standard deviations unless indicated otherwise. Between-group comparisons were conducted using a Mann–Whitney *U* test or a Kruskal–Wallis test with the Hochberg method for post hoc testing. The Shapiro–Wilk *W* test was used to determine the nature of the data distribution. Continuous variables were log-

transformed before univariate and multivariate linear regression analyses. A *P* value of <0.05 was considered significant. All statistical analyses were performed using SPSS 17.0 (SPSS, Inc., Chicago, IL, USA).

## Results

Fifty-four women in their first IVF cycle with a history of salpingectomy along with 59 age-matched controls without prior tubal disease were included in the final analysis. Among the women with a history of salpingectomy, the indication for salpingectomy was ectopic pregnancy in 26 patients and hydrosalpinx in 28 patients. As detailed in Table 1, the women with prior salpingectomy had significantly lower AFC and AMH levels than the controls without tubal disease. Age, BMI, day-3 FSH levels, total gonadotropin dose, and total stimulation days did not significantly differ between the two groups. No significant differences in age, BMI, or baseline surrogate markers of ovarian reserve and ovarian response (i.e., AMH, AFC, day-3 FSH levels, total gonadotropin dose, and total stimulation days) were observed between the patients with distinct indications of salpingectomy (Table 1).

The univariate analysis suggested that the number of retrieved oocytes was significantly and negatively associated with history of salpingectomy, age, day-3 FSH levels, and total gonadotropin dose, whereas a significant positive association was observed between the number of retrieved oocytes and AFC and AMH levels (Table 2). The aforementioned associations between the number of retrieved oocytes and the given variables remained after the model was adjusted for age and BMI. However, in the multivariate-adjusted complete model, only AFC and AMH levels continued to exhibit significant associations with the number of oocytes (Table 2).

The number of retrieved oocytes was significantly lower in patients in the study group than in those without a history of salpingectomy ($10.4 \pm 5.2$ vs. $12.2 \pm 3.8$, $P = .006$; Fig 1A). To further investigate the number of retrieved oocytes according to levels of ovarian reserve, participant AMH level and AFC were further divided into two groups to compare the numbers of retrieved oocytes in women with and without salpingectomy history and with different indications for the procedure. As indicated in Table 3 and Fig 1B, prior salpingectomy history was associated with significantly lower retrieved oocyte numbers in women with AMH < 4 ng/mL and AFC < 12, but not in women with AMH ≥ 4 ng/mL or AFC ≥ 12. Regardless of baseline

**Table 1. Basic characteristics of all enrolled patients at the time of recruitment.**

| Study group (with history of salpingectomy) | | | | | Control group (without tubal diseases) | |
|---|---|---|---|---|---|---|
| Reason for salpingectomy | Ectopic pregnancy | Hydrosalpinx | *P* value[*] | All | | *P* value[**] |
| Number | 26 | 28 | - | 54 | 59 | - |
| Age (years) | 28.6 ± 4.9 | 30.8 ± 2.6 | 0.09 | 29.7 ± 4.0 | 29.3 ± 1.8 | 0.25 |
| BMI (kg/m2) | 20.9 ± 2.5 | 20.0 ± 4.5 | 0.67 | 20.4 ± 3.7 | 20.8 ± 2.1 | 0.76 |
| AFC | 11.3 ± 7.2 | 10.4 ± 6.2 | 0.48 | 10.8 ± 6.6 | 13.2 ± 5.2 | 0.005 |
| Day3 FSH (mIU/mL) | 7.6 ± 1.8 | 7.5 ± 2.5 | 0.37 | 7.5 ± 2.2 | 7.4 ± 1.7 | 0.93 |
| AMH (ng/mL) | 5.4 ± 2.7 | 5.2 ± 3.9 | 0.44 | 5.3 ± 3.3 | 6.6 ± 3.5 | 0.039 |
| Total gonadotropin dose (IU) | 2303.8 ± 763.6 | 2528.6 ± 1051.1 | 0.59 | 2420.4 ± 922.4 | 2193.2 ± 647.3 | 0.30 |
| Total stimulation days | 9.8 ± 1.2 | 9.9 ± 1.3 | 0.77 | 9.8 ± 1.3 | 9.9 ± 1.3 | 0.59 |

BMI: body mass index; AFC: antral follicle count; AMH: anti-Müllerian hormone; FSH: follicle-stimulating hormone.

[*] Statistical comparisons between participants with distinct indications of salpingectomy.

[**] Statistical comparisons between the control and study groups.

**Table 2. Results of univariate and multivariate analysis of the association between the number of retrieved oocytes and history of salpingectomy, as well as other variables involved in predicting the ovarian response in IVF.**

| | Univariate analysis | | Multivariate analysis | | | | | |
|---|---|---|---|---|---|---|---|---|
| | | | Age adjusted | | BMI adjusted | | Full model | |
| Variables | β | P value | β | P value | β | P value | β | P value |
| **History of salpingectomy** | -0.2405 | 0.005 | -0.2353 | 0.005 | -0.2369 | 0.007 | -0.0415 | 0.56 |
| **ln(Age)** | -0.8390 | 0.004 | - | - | -0.8574 | 0.034 | -0.0053 | 0.99 |
| **ln(BMI)** | -0.1391 | 0.45 | -0.1579 | 0.38 | - | - | -0.1152 | 0.37 |
| **ln(AFC)** | 0.5887 | <0.001 | 0.5744 | <0.001 | 0.5791 | <0.001 | 0.3441 | <0.001 |
| **ln(Day3 FSH)** | -0.4541 | 0.008 | -0.5124 | 0.002 | -0.4125 | 0.018 | -0.1934 | 0.18 |
| **ln(AMH)** | 0.4335 | <0.001 | 0.4366 | <0.001 | 0.4309 | <0.001 | 0.2726 | <0.001 |
| **ln(Total gonadotropin dose)** | -0.4881 | <0.001 | -0.4688 | <0.001 | -0.4648 | <0.001 | -0.1442 | 0.39 |
| **ln(Total stimulation days)** | 0.3339 | 0.34 | 0.3461 | 0.34 | 0.3586 | 0.32 | 0.7669 | 0.021 |

BMI: body mass index; AFC: antral follicle count; AMH: anti-Müllerian hormone; FSH: follicle-stimulating hormone.

AMH level and AFC, the retrieved oocyte number did not differ significantly between patients with distinct indications of salpingectomy (i.e., ectopic pregnancy vs. hydrosalpinx; Table 3 and Fig 1C).

The women with salpingectomy history because of hydrosalpinx had a significantly lower retrieved oocyte number than the women without a history of tubal disease (Fig 2A). This association was stronger in the women with AMH < 4 ng/mL (Fig 2B), but not in those with AMH ≥ 4 ng/mL (Fig 2C). In addition, the women with unilateral salpingectomy history had

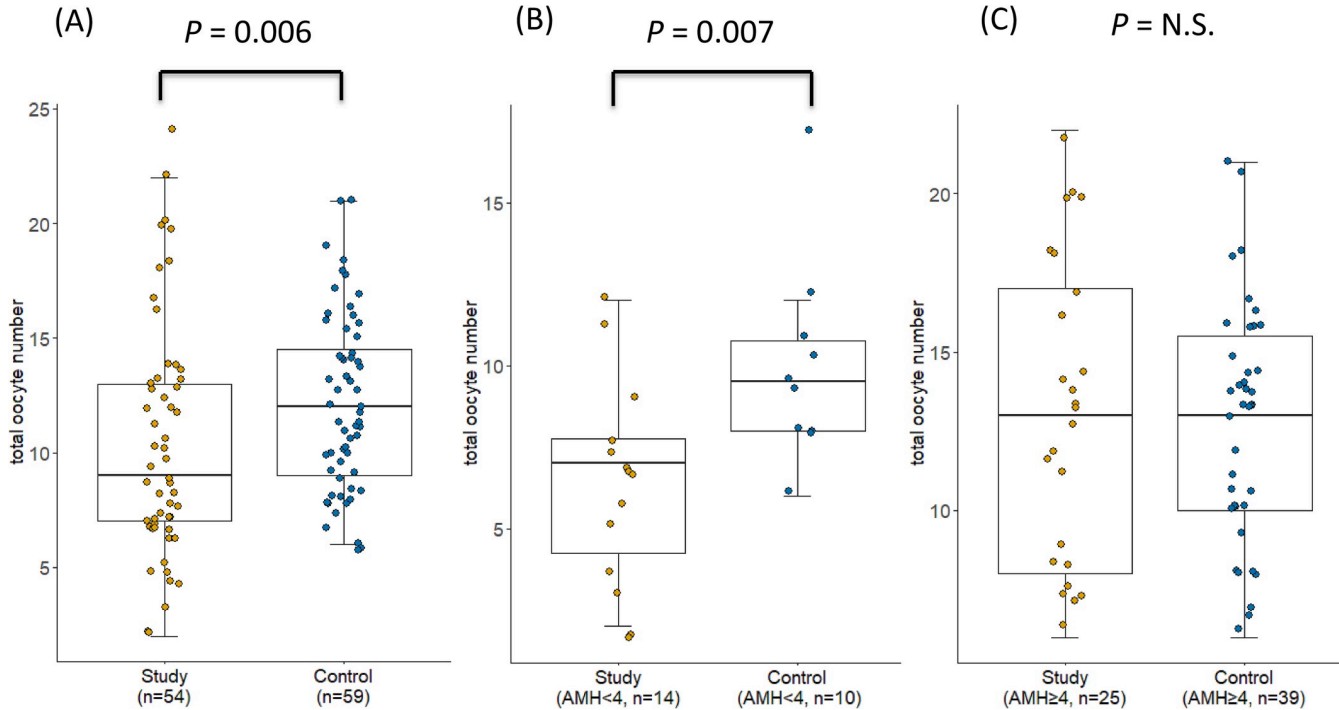

**Fig 1. Comparison of total numbers of retrieved oocytes between study and control groups.** (A) Total number of retrieved oocytes in IVF was significantly higher in the enrolled women as a whole than in those with a history of salpingectomy. Subgroup analysis of women with (B) AMH < 4 ng/mL and (C) AMH ≥ 4 ng/mL. The study and control groups comprised women with a history of salpingectomy and women without tubal disease, respectively. Values presented as mean ± SD and were provided in the minimal data set of supplementary files. Statistical analysis was conducted by Mann–Whitney *U* test. P values presented as an actual number when there was a significant difference. N.S: not significant; AMH: anti-Müllerian hormone.

**Table 3. Subgroup analysis by baseline ovarian reserve indicating the number of retrieved oocytes in women with and without salpingectomy and in women with salpingectomy and distinct indications (i.e., ectopic pregnancy vs. hydrosalpinx).**

| Study subjects (with history of salpingectomy) | | | | | Control subjects (without tubal diseases) | |
|---|---|---|---|---|---|---|
| | Ectopic pregnancy | Hydrosalpinx | P value* | All | | P value** |
| AMH < 4 (ng/mL) | | | | | | |
| Oocyte number | 6.9 ± 2.9 | 6.0 ± 3.4 | 0.75 | 6.4 ± 3.1 | 9.9 ± 3.0 | 0.007 |
| AMH ≥ 4 (ng/mL) | | | | | | |
| Oocyte number | 12.8 ± 3.7 | 13.3 ± 5.8 | 0.96 | 13.1 ± 4.8 | 12.8 ± 3.7 | 0.52 |
| AFC < 12 | | | | | | |
| Oocyte number | 8.2 ± 2.9 | 8.0 ± 5.2 | 0.31 | 8.1 ± 4.4 | 10.2 ± 3.1 | 0.004 |
| AFC ≥ 12 | | | | | | |
| Oocyte number | 12.8 ± 3.5 | 15.2 ± 5.3 | 0.30 | 13.8 ± 4.4 | 13.4 ± 3.7 | 0.56 |

AFC: antral follicle count; AMH: anti-Müllerian hormone.

* Statistical comparisons between participants with distinct indications of salpingectomy.

** Statistical comparisons between the control and study groups.

a significantly lower number of retrieved oocytes than the women without tubal disease (Fig 3A). Furthermore, this association was significant in women with AMH < 4 ng/mL (Fig 3B) but not in those with AMH ≥ 4 ng/mL (Fig 3C).

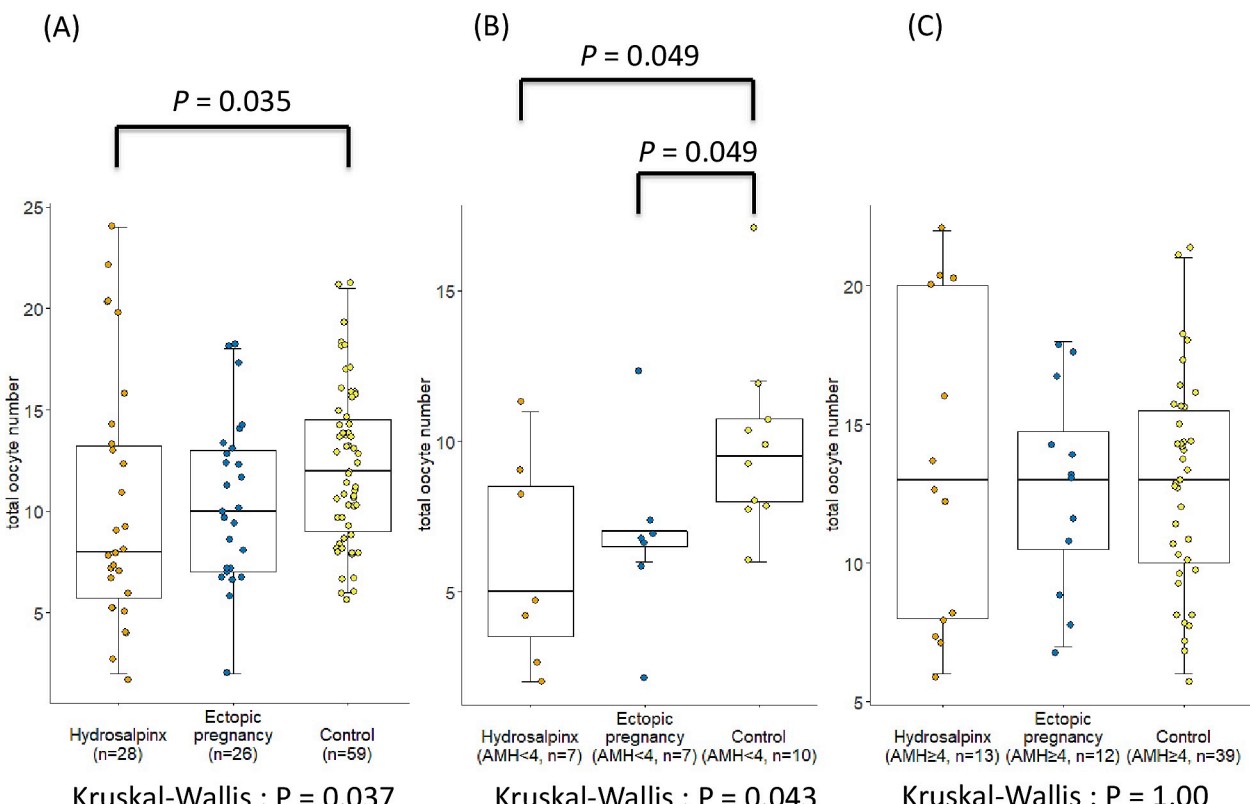

**Fig 2. Total number of retrieved oocytes in women with distinct indications of salpingectomy.** Comparison of the differences in the numbers of retrieved oocytes between women without tubal disease (control) and women with different indications for prior salpingectomy (i.e., hydrosalpinx and ectopic pregnancy) (A) in all enrolled participants, (B) in women with AMH < 4 ng/mL, and (C) in women with AMH ≥ 4 ng/mL. Values were provided in the minimal data set of supplementary files. Statistical analysis was by Kruskal–Wallis test with the Hochberg post hoc testing. AMH: anti-Müllerian hormone.

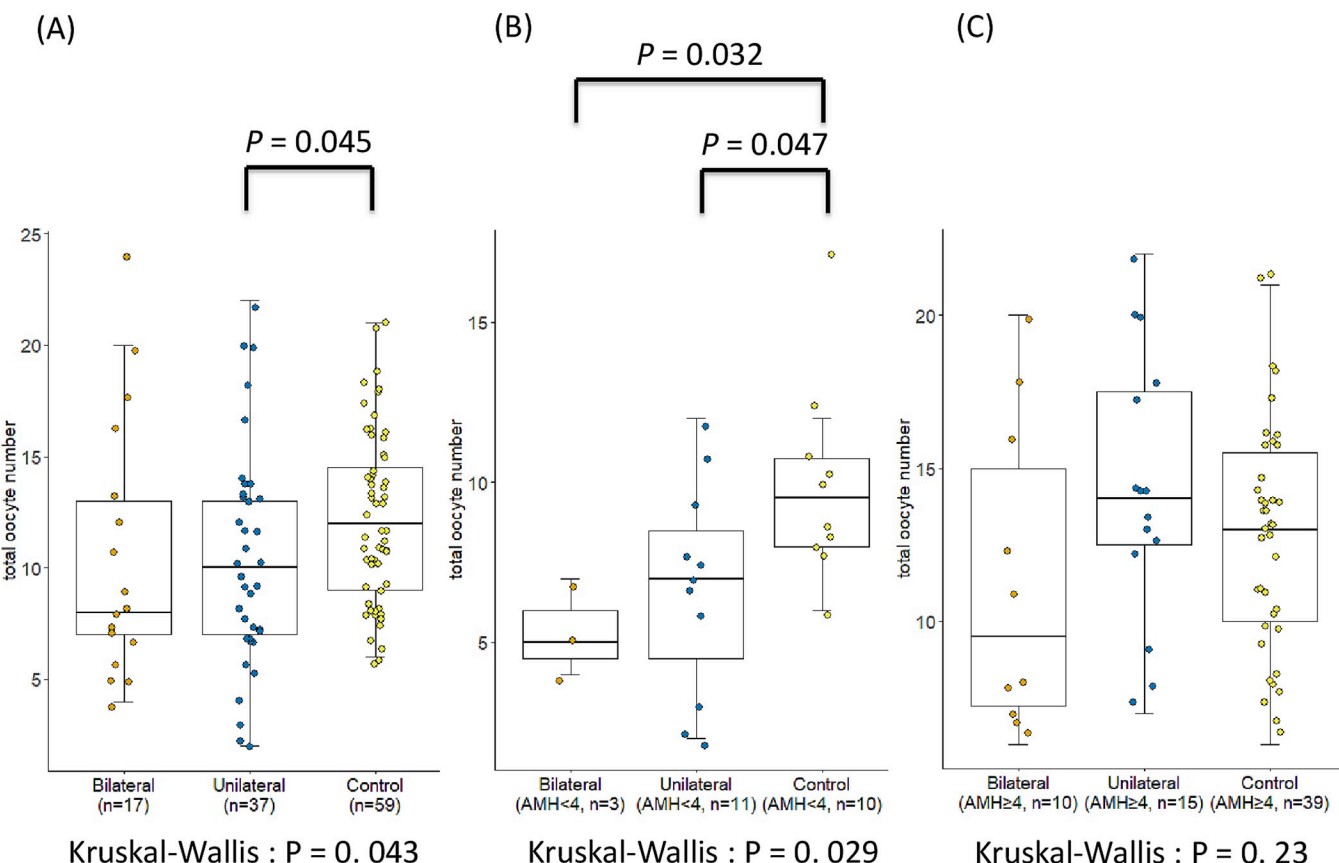

**Fig 3. Total numbers of retrieved oocytes in those with bilateral, unilateral, and without salpingectomy histories.** Differences in the retrieved oocyte numbers in women with bilateral salpingectomy, unilateral salpingectomy, and without salpingectomy (A) in all enrolled subjects, (B) in women with AMH < 4 ng/mL, and (C) in women with AMH ≥ 4 ng/mL. AMH: anti-Müllerian hormone.

## Discussion

In this study, the number of retrieved oocytes in young patients with a history of salpingectomy who were undergoing IVF was significantly lower than that in the controls without tubal disease. In addition, the subgroup analysis revealed that the reduction in the retrieved oocyte number related to prior salpingectomy was prominent in patients without optimal baseline ovarian reserve. This study investigated the effect of prior salpingectomy and its indications on the subsequent ovarian response in women undergoing IVF. To our knowledge, this is the first study of such topics that specifically focused on patients younger than 35-year-old who received a GnRH antagonist protocol for controlled ovarian stimulation, compared the study population with a control group that did not have underlying tubal disease, and examined the effect of baseline ovarian reserve.

Several studies have reported a beneficial effect of salpingectomy on embryo implantation and clinical pregnancy rate in women with infertility and hydrosalpinges [7–9]. However, the potential damage in the vascular perfusion of the ovary due to salpingectomy is concerning [2]. In a randomized controlled trial, proximal tubal obstruction outperformed salpingectomy for preserving ovarian reserve in patients with hydrosalpinges, further demonstrating the potentially harmful effect of aggressive salpingectomy on this reserve [17]. Nevertheless, salpingectomy procedures are becoming increasingly common for various conditions in young women who undergo gynecologic surgery to avoid persistent disease or reduce the risk of

future ovarian malignancy. Therefore, a thorough understanding of whether this surgical intervention affects the subsequent ovarian reserve and ovarian response is warranted to support future fertility and medical considerations.

Previous research has reported that a longer duration of ovarian stimulation and a higher dose of gonadotropin was required for IVF in women who underwent salpingectomy than in those who did not [22], especially for those in which ectopic pregnancy is the indication [12]. Most studies, including meta-analyses, have reported no significant differences in terms of ovarian reserve (represented by AMH, AFC, and day-2 or day-3 FSH levels) or the ovarian response (represented by the number of oocytes retrieved, duration of ovarian stimulation, and dose of gonadotropin in IVF before and after salpingectomy) [2, 8, 9, 11, 12, 15, 23], regardless of the indication of salpingectomy [15, 23]. However, these studies have mainly compared the ovarian reserve before and after surgery and were limited to populations with underlying tubal disease and middle-aged women. Therefore, the negative effect of salpingectomy on ovarian function may not have been detectable in these previous study populations.

The present study demonstrates that women without an optimal baseline ovarian reserve may be more vulnerable to damage from salpingectomy. For women with an optimal ovarian reserve and a large follicle pool, a history of salpingectomy did not affect the number of retrieved oocytes in subsequent IVF cycles. Similar results were observed in a retrospective study [13], which revealed that in IVF patients aged 35–39 years, the AFC was lower in patients with a history of salpingectomy than in those without a history of salpingectomy; however, the study did not further stratify patients under the age of 35 with distinct baseline ovarian reserves. Moreover, in the study of Chen et al., the researchers selected patients with tubal disease as controls, which may have lessened the effect of salpingectomy, for underlying tubal disease may have been a confounding factor for reduced ovarian reserve. Therefore, this may be unable to demonstrate the difference in ovarian response for patients with salpingectomy history and a sufficient baseline ovarian reserve or those under 35 years old. Nonetheless, these findings substantiate the negative effect of salpingectomy on women with suboptimal baseline ovarian reserve; therefore, baseline ovarian reserves should be considered in treatment strategy recommendations for women.

The primary outcome in our study was actual retrieved oocyte number. We contend that to obtain more embryos, using retrieved oocyte number as a reference would be superior to using circulating surrogate markers of ovarian reserve. The number of retrieved oocytes was also positively correlated with pregnancy outcome. The number of retrieved oocytes in relation to the total dose of ovarian stimulation drugs could reflect the ovarian response. However, a retrospective study [24] reported a negative effect of salpingectomy history on AMH levels but not on the retrieved oocyte number for women with infertility, which differs from our study results. Nevertheless, distinct from our study, short and minimal stimulation protocols were applied for controlled ovarian hyperstimulation in that retrospective study. In our study, we solely used the GnRH antagonist protocol for controlled ovarian stimulation and adjusted the dosage to obtain the optimal response. Variations in stimulation protocol, total gonadotropin dosage, stimulation interval, baseline ovarian reserve, and blood flow in the ovaries may have influenced the numbers of retrieved oocytes.

Along with concern about damage of periovarian vascular perfusion and potential thermal damage by an electrocauterization device during salpingectomy, underlying fallopian tube diseases that lead to hydrosalpinges and ectopic pregnancy may also contribute to impaired ovarian response. Pelvic inflammatory disease and mild peritoneal endometriosis are common causes of tubal adhesion or malfunction. In patients without prior surgical intervention, chronic pelvic inflammation and tubal obstruction reportedly reduce ovarian reserves, as indicated by lower AMH levels in these patients than in healthy women with infertility [25]. In

addition, direct damage to the gonads by iron deposition has been demonstrated in human and animal studies [26–28]. A recent meta-analysis also revealed a reduced ovarian reserve in women with endometriosis, as indicated by lower AFC and AMH levels, regardless of the presence of endometriotic ovarian cysts [29]. However, there were several limitations in our study. Because of limited case numbers and our retrospective study design, we did not consider tubal disease itself as a potential confounding factor affecting ovarian response. Untreated hydrosalpinges can also hinder the procedure of transvaginal oocyte retrieval and increase the risk of infection. In addition, when considering salpingectomy's effect on ovarian response, designing the study to compare retrieved oocyte numbers before and after tubal surgery would not be feasible in the ectopic group because of the emergency nature of the disease. Also, the univariate analysis in our study indicated that factors such as salpingectomy history, AMH, and AFC were all significantly associated with the number of retrieved oocytes. However, the AMH and AFC had more direct and stronger correlations with the retrieved oocyte number than did history of salpingectomy, the effect of salpingectomy history was no longer significant after applying the full model of multivariate analysis. In conclusion, compared with age-matched women without tubal disease, young women with a history of salpingectomy (with either ectopic pregnancy or hydrosalpinx indications) had a significantly lower number of retrieved oocytes during IVF, and this was especially evident in women without a large ovarian follicle pool. Our study findings suggest that greater attention should be directed toward ovarian response in women with infertility, without sufficient ovarian reserves, and with a history of salpingectomy because of their vulnerability to salpingectomy-related damage.

## Supporting information

**S1 File.**
(DOCX)

## Author Contributions

**Conceptualization:** Cheng-Yu Ho, Yu-Hung Lin, Mei-Jou Chen.

**Data curation:** Cheng-Yu Ho, Yu-Hung Lin.

**Formal analysis:** Yu-Yuan Chang, Mei-Jou Chen.

**Investigation:** Cheng-Yu Ho.

**Methodology:** Cheng-Yu Ho, Yu-Yuan Chang, Mei-Jou Chen.

**Software:** Yu-Yuan Chang.

**Supervision:** Mei-Jou Chen.

**Writing – original draft:** Cheng-Yu Ho, Mei-Jou Chen.

**Writing – review & editing:** Mei-Jou Chen.

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
