## [Decision Letter · Decision Letter 0]

24 Nov 2021

PONE-D-21-34495

Prior Salpingectomy Impairs the Retrieved Oocyte Number in In Vitro Fertilization Cycles of Women Under 35 Years Old Without Optimal Ovarian Reserve :  a retrospective study

PLOS ONE

Dear Dr. Ho,

Thank you for submitting your manuscript to PLOS ONE. After careful consideration, we feel that it has merit but does not fully meet PLOS ONE’s publication criteria as it currently stands. Therefore, we invite you to submit a revised version of the manuscript that addresses the points raised during the review process.

ACADEMIC EDITOR: This study investigating the association between the history of salpingectomy and the number of oocytes in women undergoing IVF has some issues that need to be addressed. Specifically, 2. the English needs a revision; 2. The authors should convince the reader that there were no selection biases; 3. Limits of the study needs to be discussed; 4. Several comparisons were performed. Please perform a Bonferroni's correction. All the issues raised by the Reviewers should be carefully addressed.

We look forward to receiving your revised manuscript.

Kind regards,

Paola Viganò

Academic Editor

PLOS ONE

Journal Requirements:

5. Please amend either the title on the online submission form (via Edit Submission) or the title in the manuscript so that they are identical.

Additional Editor Comments:

This study investigating the association between the history of salpingectomy and the number of oocytes in women undergoing IVF has some issues that need to be addressed. Specifically, 2. the English needs a revision; 2. The authors should convince the reader that there were no selection biases; 3. Limits of the study needs to be discussed; 4. Several comparisons were performed. Please perform a Bonferroni correction. All the issues raised by the Reviewers should be carefully addressed.

Reviewers' comments:

Reviewer's Responses to Questions

Comments to the Author

1. Is the manuscript technically sound, and do the data support the conclusions?

Reviewer #1: Yes

Reviewer #2: Yes

2. Has the statistical analysis been performed appropriately and rigorously?

Reviewer #1: No

Reviewer #2: Yes

3. Have the authors made all data underlying the findings in their manuscript fully available?

Reviewer #1: Yes

Reviewer #2: Yes

4. Is the manuscript presented in an intelligible fashion and written in standard English?

Reviewer #1: Yes

Reviewer #2: Yes

5. Review Comments to the Author

Reviewer #1: The authors investigated the association between the history of salpingectomy and the number of oocytes in young women undergoing IVF. They found that the number of retrieved oocytes was significantly lower in patients with a history of salpingectomy, and the influence was more evident in women with AMH<4.

The paper provides important data, but it still needs some revision to be acceptable for the PLOS ONE.

#1 It is unclear why the author investigated women under 35 years old. Ref No.13 showed no influence of salpingectomy on AFC in women under 35. Please clarify in the Introduction.

#2 A previous retrospective study (shown below) also showed the influence of salpingectomy on ovarian reserve in IVF-ET patients. The study showed a negative effect on AMH but no impact on retrieved oocytes. The number of retrieved oocytes was the primary outcome of the present study. Please discuss the impact on retrieved oocytes at great length in the Discussion.

1. Ye XP, Yang YZ, Sun XX. A retrospective analysis of the effect of salpingectomy on serum antiMullerian hormone level and ovarian reserve. American journal of obstetrics and gynecology. 2015;212(1):53 e1-10.

#3 In Table 2, the history of salpingectomy is not statistically significant in full model multivariate analysis. The author should explain this point.

#4 The author should explain the limitation of this study. The result was not conclusive. There are some limitations regarding retrospective design, sample size, or baseline difference between two groups. As mentioned in the Discussion, it is also possible that the baseline difference of AFC and AMH was not the result of salpingectomy but other pelvic conditions.

#5 In figure 2 and 3, p-values of Kruskal–Wallis tests are needed. The correction method should be shown to compare groups (e.g., Bonferroni correction).

#6 A significant figures of p-values are not the same. (Figure 1A p=0.006, Figure 2 and 3 p<0.05, and p<0.01 are used in tables)

#7 The author should decrease tone, especially in conclusion, because this study has several limitations.

Reviewer #2: Dear Authors,

The subject of the study is current. A clear, simple and understandable writing language was used in the article. I believe it will benefit the reader. This is a good study comparing the number of oocytes collected after salpingectomy. Congratulations to the authors.

Below are my recommendations for the study.

Best regards

- Some english grammar mistakes should be corrected.

-I suggest that some studies comparing the histopathological effects of salpingectomy and proximal tubal occlusion procedures on ovarian tissue should be given as examples [e.g. Atilgan R, Pala Ş, Kuloğlu T, Şanli C, Yavuzkir Ş, Özkan ZS. Comparison of the efficacy between bilateral proximal tubal occlusion and total salpingectomy on ovarian reserve and the cholinergic system: an experimental study. Turk J Med Sci. 2020 Jun 23;50(4):1097-1105. doi: 10.3906/sag-2002-179. ..................

Atilgan R, Kuloğlu T, Boztosun A, Orak U, Baspinar M, Can B, Sapmaz E. Investigation of the effects of unilateral total salpingectomy on ovarian proliferating cell nuclear antigen and follicular reserve: experimental study. Eur J Obstet Gynecol Reprod Biol. 2015 May;188:56-60. doi: 10.1016/j.ejogrb.2015.02.028.]

6. PLOS authors have the option to publish the peer review history of their article (what does this mean?). If published, this will include your full peer review and any attached files.

Do you want your identity to be public for this peer review? For information about this choice, including consent withdrawal, please see our Privacy Policy.

Reviewer #1: No

Reviewer #2: Yes: Remzi Atılgan

---

## [Author Response · Author response to Decision Letter 0]

18 Jan 2022

Comment Responses and Rebuttal

We greatly appreciated the editor and reviewers’ valuable comments. We have revised our manuscript accordingly and for adherence to the journal requirements. Our point-by-point responses to your comments are as follows.

# Additional Editor Comments

This study investigating the association between the history of salpingectomy and the number of oocytes in women undergoing IVF has some issues that need to be addressed. Specifically, 

Comment 1. The English needs a revision. 

Response 1: We apologize for the grammar mistakes in our manuscript. We have commissioned the help of an English-language editing team to revise our manuscript. We have attached the certificate for this service as a supplementary file.

Comment 2. The authors should convince the reader that there were no selection biases.

Response 2: In this study conducted from January 2012 through December 2019, all the women with infertility who met the inclusion and exclusion criteria of the study underwent oocyte retrieval at the Department of Obstetrics and Gynecology at Shin Kong Wu Huo Shih Hospital and were enrolled consecutively for analysis. The age-matched control subjects were enrolled by 3-year age strata, with the first names on the lists selected. No patients were excluded from final analysis as outliers. We have avoided selection bias as much as possible by employing these methods, which we have further detailed in the “Materials and methods” section. 

Comment 3. Limits of the study needs to be discussed. 

Response 3: This study has several limitations, including its retrospective design, small sample size, and lack of measurement before salpingectomy. In addition, because we only selected women without underlying tubal disease as our controls, those women with tubal disease but without tubal surgery history were not considered in this study. We have addressed these limitations in our revised manuscript in the revised “Discussion” section, page 19 and 20, lines 388–394. 

Comment 4. Several comparisons were performed. Please perform a Bonferroni correction. 

Response 4: Because the Bonferroni test is reportedly too stringent to identify the between-group differences with such a small sample size, we reanalyzed the data by applying nonparametric Kruskal–Wallis test, with Hochberg method for post hoc testing instead; we have described these procedures in our revised “Statistical Analysis” section and in the revised versions of Figs 2 and 3. 

# Reviewer 1’ s comments:

The authors investigated the association between the history of salpingectomy and the number of oocytes in young women undergoing IVF. They found that the number of retrieved oocytes was significantly lower in patients with a history of salpingectomy, and the influence was more evident in women with AMH<4.

The paper provides important data, but it still needs some revision to be acceptable for the PLOS ONE.

Comment 1. It is unclear why the author investigated women under 35 years old. Ref No.13 showed no influence of salpingectomy on AFC in women under 35. Please clarify in the Introduction.

Response 1: 

Thank you for your comments and suggestions. In the study of Chen et al. (i.e., Ref 13), a lower AFC was only discovered in women with infertility who were aged 35–39 years in response to salpingectomy and not in those women under the age of 35 years. This may imply an age-related ovarian function decline that highlights the negative effect of salpingectomy on ovarian reserves. However, Chen et al. did not perform subgroup analyses to investigate whether the effect of salpingectomy was also more pronounced in women aged under 35 years with suboptimal ovarian reserves. Moreover, in the study of Chen et al., the researchers selected patients with tubal disease as controls; the effect of salpingectomy may thus have been reduced, with underlying disease possibly acting as a confounding factor that reduced ovarian reserves. We have further clarified this matter in the revised “Introduction” on page 6, lines 124–131). 

Comment 2. A previous retrospective study (shown below) also showed the influence of salpingectomy on ovarian reserve in IVF-ET patients. The study showed a negative effect on AMH but no impact on retrieved oocytes. The number of retrieved oocytes was the primary outcome of the present study. Please discuss the impact on retrieved oocytes at great length in the Discussion.

1. Ye XP, Yang YZ, Sun XX. A retrospective analysis of the effect of salpingectomy on serum anti-Mullerian hormone level and ovarian reserve. American journal of obstetrics and gynecology. 2015;212(1):53 e1-10.

Response 2: 

We have cited the reference as [24] and discussed it in the “Discussion” section on pages 18 and 19, lines 364–376 in our revised manuscript as follows: 

The primary outcome in our study was actual retrieved oocyte number. We contend that to obtain more embryos, using retrieved oocyte number as a reference would be superior to using circulating surrogate markers of ovarian reserve. The number of retrieved oocytes was also positively correlated with pregnancy outcome. The number of retrieved oocytes in relation to the total dose of ovarian stimulation drugs could reflect the ovarian response. However, a retrospective study [24] reported a negative effect of salpingectomy history on AMH levels but not on the retrieved oocyte number for women with infertility, which differs from our study results. Nevertheless, distinct from our study, short and minimal stimulation protocols were applied for controlled ovarian hyperstimulation in that retrospective study. In the present study, we solely used the GnRH antagonist protocol for controlled ovarian stimulation and adjusted the dosage to obtain the optimal response. Variations in stimulation protocol, total gonadotropin dosage, stimulation interval, baseline ovarian reserve, and (potentially) blood flow in the ovaries may have influenced the numbers of retrieved oocytes.

Comment 3. In Table 2, the history of salpingectomy is not statistically significant in full model multivariate analysis. The author should explain this point.

Response 3: 

The univariate analysis indicated that factors such as salpingectomy history, AMH, and AFC were all significantly associated with the number of retrieved oocytes. However, because the AMH and AFC had more direct and stronger correlations with the retrieved oocyte number than did history of salpingectomy, the effect of salpingectomy history was no longer significant after applying the full model of multivariate analysis that included adjustment for AMH level and AFC. This has been described on page 13 of the revised manuscript, lines 235–241.

Comment 4. The author should explain the limitation of this study. The result was not conclusive. There are some limitations regarding retrospective design, sample size, or baseline difference between two groups. As mentioned in the Discussion, it is also possible that the baseline difference of AFC and AMH was not the result of salpingectomy but other pelvic conditions.

Response 4: 

 This study has several limitations, including the retrospective study design, small sample size, and lack of measurement before salpingectomy. In addition, because we only selected women without underlying tubal disease as controls, those women with tubal disease but without tubal surgery history were not considered in this study. We have addressed these limitations in our revised manuscript in the “Discussion” section, pages 19 and 20, lines 388–394.

Comment 5. In figure 2 and 3, p-values of Kruskal–Wallis tests are needed. The correction method should be shown to compare groups (e.g., Bonferroni correction). 

Response 5: 

In accordance with your suggestion, we have analyzed the data by applying a nonparametric Kruskal–Wallis test for the data in Figs 2 and 3. However, because the Bonferroni test is reportedly too stringent to identify the between-group differences with such a small sample size, we used the Hochberg correction method for post hoc testing instead. We have described these procedures in the revised “Statistical Analysis” section and illustrated the results in the revised Figs 2 and 3.

Comment 6. Significant figures of p-values are not the same. (Figure 1A p=0.006, Figure 2 and 3 p<0.05, and p<0.01 are used in tables).

Response 6: 

We have amended all P-value notation in our revised manuscript in accordance with the conventional mode of representation. If P values are between .9 and .001, the actual number is presented. P values of >.9 or <.001 are presented as “>.9” or “<.001.”

Comment 7. The author should decrease tone, especially in conclusion, because this study has several limitations.

Response 7:

We have rewritten the conclusion (in the last paragraph of the “Discussion” section) in our revised manuscript. Rather than the original stronger recommendation, we have changed the sentence to suggest directing greater attention toward ovarian response in women who underwent IVF with infertility, without sufficient ovarian reserves, and with salpingectomy history.

# Reviewer 2’ s comments: Dear Authors,

The subject of the study is current. A clear, simple and understandable writing language was used in the article. I believe it will benefit the reader. This is a good study comparing the number of oocytes collected after salpingectomy. Congratulations to the authors.

Below are my recommendations for the study.

Best regards

Comment 1. Some English grammar mistakes should be corrected. 

Response 1: 

Thank you for your valuable comments and suggestions. We apologize for the sporadic grammar mistakes in our manuscript. We have revised our manuscript with the help of an English-language editing service and have attached the relevant certificate as a supplementary file.

Comment 2. I suggest that some studies comparing the histopathological effects of salpingectomy and proximal tubal occlusion procedures on ovarian tissue should be given as examples [e.g. Atilgan R, Pala Ş, Kuloğlu T, Şanli C, Yavuzkir Ş, Özkan ZS. Comparison of the efficacy between bilateral proximal tubal occlusion and total salpingectomy on ovarian reserve and the cholinergic system: an experimental study. Turk J Med Sci. 2020 Jun 23;50(4):1097-1105. doi: 10.3906/sag-2002-179. ..................

Atilgan R, Kuloğlu T, Boztosun A, Orak U, Baspinar M, Can B, Sapmaz E. Investigation of the effects of unilateral total salpingectomy on ovarian proliferating cell nuclear antigen and follicular reserve: experimental study. Eur J Obstet Gynecol Reprod Biol. 2015 May;188:56-60. doi: 10.1016/j.ejogrb.2015.02.028.]

Response 2:

We have cited these references as [18] and [19] and used the studies as examples in the “Introduction” section on page 6, lines 112–116 in our revised manuscript as follows:

An experimental study revealed that total salpingectomy in rats leads to more significant damage than proximal tubal occlusion in ovarian histopathology and the cholinergic system [18]. Another study also indicated that the unilateral total salpingectomy procedure can be detrimental to ipsilateral ovarian tissue as a result of ischemia–reperfusion injury [19].

---

## [Decision Letter · Decision Letter 1]

16 Feb 2022

PONE-D-21-34495R1Prior salpingectomy impairs the retrieved oocyte number in in vitro fertilization cycles of women under 35 years old without optimal ovarian reservePLOS ONE

Dear Dr. Chen,

Thank you for submitting your manuscript to PLOS ONE. After careful consideration, we feel that it has merit but does not fully meet PLOS ONE’s publication criteria as it currently stands. Therefore, we invite you to submit a revised version of the manuscript that addresses the points raised during the review process.

ACADEMIC EDITOR: 

Although the paper has been improved, there are still some problems that need to be addressed:

The Authors stated in their response ‘the effect of salpingectomy history was no longer significant after applying the full model of multivariate analysis that included adjustment for AMH level and AFC’ This is a strong message that should be clear in the Abstract.The abstract needs to be improved as the message provided is misleading.Figure legends are missing. How are data presented in the bars?The choice of limits of AMH (4 ng/mL) and AFC (12) for the subgroup analysis needs to be justified.==============================

We look forward to receiving your revised manuscript.

Kind regards,

Paola Viganò

Academic Editor

PLOS ONE

Journal Requirements:

Additional Editor Comments:

Although the paper has been improved, there are still some problems that need to be addressed:

1. The Authors stated in their response ‘the effect of salpingectomy history was no longer significant after applying the full model of multivariate analysis that included adjustment for AMH level and AFC’ This is a strong message that should be clear in the Abstract.

2. The abstract needs to be improved as the message provided is misleading.

3. Figure legends are missing. How are data presented in the bars?

4. The choice of limits of AMH (4 ng/mL) and AFC (12) for the subgroup analysis needs to be justified.

Reviewers' comments:

Reviewer's Responses to Questions

**Comments to the Author**

1. If the authors have adequately addressed your comments raised in a previous round of review and you feel that this manuscript is now acceptable for publication, you may indicate that here to bypass the “Comments to the Author” section, enter your conflict of interest statement in the “Confidential to Editor” section, and submit your "Accept" recommendation.

Reviewer #1: All comments have been addressed

Reviewer #2: All comments have been addressed

2. Is the manuscript technically sound, and do the data support the conclusions?

Reviewer #1: Yes

Reviewer #2: Yes

3. Has the statistical analysis been performed appropriately and rigorously? 

Reviewer #1: Yes

Reviewer #2: Yes

4. Have the authors made all data underlying the findings in their manuscript fully available?

Reviewer #1: Yes

Reviewer #2: Yes

5. Is the manuscript presented in an intelligible fashion and written in standard English?

Reviewer #1: Yes

Reviewer #2: Yes

6. Review Comments to the Author

Reviewer #1: Thank you for your patient revision.

The manuscript has been revised well.

I think this manuscript will be acceptable for PLOS ONE.

Reviewer #2: The subject of the manuscript is current and I think it will be useful to the reader. The authors made the necessary corrections in line with the suggestions. I think the article can be published in its current form. Congratulations to the authors.

7. PLOS authors have the option to publish the peer review history of their article (what does this mean?). If published, this will include your full peer review and any attached files.

Reviewer #1: **Yes: **Tsutomu IDA

Reviewer #2: **Yes: **Remzi Atılgan

---

## [Author Response · Author response to Decision Letter 1]

11 Apr 2022

Dear Academic editor: 

We greatly appreciated the valuable comments for improving our study. We have revised our manuscript accordingly and for adherence to the journal requirements. Our point-by-point responses to your comments are as follows.

# Academic Editor Comments

Although the paper has been improved, there are still some problems that need to be addressed: 

Comment 1. The Authors stated in their response ‘the effect of salpingectomy history was no longer significant after applying the full model of multivariate analysis that included adjustment for AMH level and AFC’ This is a strong message that should be clear in the Abstract.

Response 1: Indeed, this message should be clear, and we have added in the Abstract, subtitled “measurements and main results”, line 64-69. Also, we have modified the context of the “conclusion” section in the Abstract with highlighting the message as stated above. 

The problem that univariate regression analysis in our study indicated that factors such as salpingectomy history, AMH, and AFC were all significantly associated with the number of retrieved oocytes. However, in the multivariate-adjusted complete model, only AFC and AMH levels continued to exhibit significant associations with the number of retrieved oocytes. The explanation is that the effect of previous salpingectomy toward ovarian response are not as direct as ovarian surrogate biomarkers, which had been collected just around the time of controlled ovarian stimulation and transvaginal oocytes retrieval. 

Salpingectomy impaired the circulation of ovary and may cause ischemic-reperfusion injury, and indirectly affect the ovarian function and response to fertility drugs afterwards. Therefore, in the full model of multivariate analysis, we can see that AFC and AMH have more direct influences toward number of retrieved oocytes than other factors. 

Comment 2. The abstract needs to be improved as the message provided is misleading.

Response 2: Thank you for the comments and suggestions. We have added the important message of our statistical results in the Abstract, subtitled” measurements and main results” and “conclusion”. Although the full mode of multivariate analysis did not reveal the negative impact of previous salpingectomy on the number of retrieved oocytes, the detrimental effect to the ovarian function and response still presented via the decreased ovarian surrogate markers (AFC and AMH). Therefore, infertile patients with previous salpingectomies and suboptimal ovarian reserve should be paid attention to their ovarian response, however, postsurgical AMH and AFC levels still possess a more direct predictive value on ovarian response than the history of salpingectomy. 

Comment 3. Figure legends are missing. How are data presented in the bars?

Response 3: We do apologize for the mistakes of incomplete figure legends. According to the policy of PLOS ONE journal for figure legends, we inserted the figure legends (including Fig. 1 to 3) by the way of inlay in the Manuscript, page 14 and 16. The data in the bars, including the values of means, standard deviations, were provided in the minimal data set of supplementary files. P values presented as an actual number in the figure above the bar plot when there was a significant difference. 

Comment 4. The choice of limits of AMH (4 ng/mL) and AFC (12) for the subgroup analysis needs to be justified.

Response 4: In the full mode of multivariate analysis within our study, we found that AMH and AFC continued to be significantly associated with the number of retrieved oocytes. We further divided the subgroups with different level of surrogate ovarian biomarkers to realize the impact on ovarian response. 

The cutoff value for AMH (4 ng/mL) was based on the reported 75th percentile of AMH levels of generally healthy Chinese women aged 35 years by the reference [21], BJOG 2020;127(6):720-8. The 75th percentile of AMH levels between 35 to 36 year-old chinese women lies between 4.229 to 3.973 ng/mL, which can be found in the Table 1 of the reference. Another reason for the cutoff value is because the limited case number in our study, we chose accordingly rather than calculated median AMH of our study. The cutoff value for AFC (12 follicles) was based on the calculated median AFC of all enrolled patients in our study. Above description was written in the Materials and Methods, section “Controlled ovarian stimulation and outcome measures” last two sentences. 

The reason we used subgroup analysis to distinguish the patients with optimal ovarian reserve, which can be normal or hyper-responders and often with plenty number of retrieved oocytes and could mask the effect of history of salpingectomy.

---

## [Editor Report · Decision Letter 2]

21 Apr 2022

Prior salpingectomy impairs the retrieved oocyte number in in vitro fertilization cycles of women under 35 years old without optimal ovarian reserve

PONE-D-21-34495R2

Dear Dr. Chen,

We’re pleased to inform you that your manuscript has been judged scientifically suitable for publication and will be formally accepted for publication once it meets all outstanding technical requirements.

Kind regards,

Paola Viganò

Academic Editor

PLOS ONE
---

## [Editor Report · Acceptance letter]

25 Apr 2022

PONE-D-21-34495R2 

Prior salpingectomy impairs the retrieved oocyte number in *in vitro* fertilization cycles of women under 35 years old without optimal ovarian reserve 

Dear Dr. Chen:

I'm pleased to inform you that your manuscript has been deemed suitable for publication in PLOS ONE. Congratulations! Your manuscript is now with our production department. 

Kind regards, 

on behalf of

Dr. Paola Viganò 

Academic Editor

PLOS ONE